# Application of Digital Twins and Metaverse in the Field of Fluid Machinery Pumps and Fans: A Review

**DOI:** 10.3390/s22239294

**Published:** 2022-11-29

**Authors:** Bin Yang, Shuang Yang, Zhihan Lv, Faming Wang, Thomas Olofsson

**Affiliations:** 1School of Energy and Safety Engineering, Tianjin Chengjian University, Tianjin 300384, China; 2Department of Applied Physics and Electronics, Umeå University, SE-90187 Umeå, Sweden; 3Department of Game Design, Faculty of Arts, Uppsala University, SE-75105 Uppsala, Sweden; 4Department of Biosystems, KU Leuven, BE-3001 Leuven, Belgium

**Keywords:** digital twins, Metaverse fluid machinery, fan, pump

## Abstract

Digital twins technology (DTT) is an application framework with breakthrough rules. With the deep integration of the virtual information world and physical space, it becomes the basis for realizing intelligent machining production lines, which is of great significance to intelligent processing in industrial manufacturing. This review aims to study the application of DTT and the Metaverse in fluid machinery in the past 5 years by summarizing the application status of pumps and fans in fluid machinery from the perspective of DTT and the Metaverse through the collection, classification, and summary of relevant literature in the past 5 years. The research found that in addition to relatively mature applications in intelligent manufacturing, DTT and Metaverse technologies play a critical role in the development of new pump products and technologies and are widely used in numerical simulation and fault detection in fluid machinery for various pumps and other fields. Among fan-type fluid machinery, twin fans can comprehensively use technologies, such as perception, calculation, modeling, and deep learning, to provide efficient smart solutions for fan operation detection, power generation visualization, production monitoring, and operation monitoring. Still, there are some limitations. For example, real-time and accuracy cannot fully meet the requirements in the mechanical environment with high-precision requirements. However, there are also some solutions that have achieved good results. For instance, it is possible to achieve significant noise reduction and better aerodynamic performance of the axial fan by improving the sawtooth parameters of the fan and rearranging the sawtooth area. However, there are few application cases of the Metaverse in fluid machinery. The cases are limited to operating real equipment from a virtual environment and require the combination of virtual reality and DTT. The application effect still needs further verification.

## 1. Introduction

Fluid machinery has wide applications, such as municipal water conservancy. It can also be used in the petrochemical industry, agriculture, electric power, national defense, and other fields [1]. Fluid machinery provides power supply for the rapid development of China’s economy and science and technology [2]. However, its wide range of applications follows high energy consumption (EC) problems, consuming over 30% of the total power generation in China. At the same time, low operating efficiency has also been noted in fluid machinery [3]. For example, the fan often deviates from the design working condition in actual operations, leading to the decline of both the fan’s efficiency and the overall operating efficiency. The result is low energy utilization (EU). The relative energy shortages are hindering China’s energy-intensive industries’ development, which is becoming increasingly serious. To reduce EC and improve EU, higher requirements are put forward for the design and operation of fluid machinery [4]. The centrifugal fan is a universal fluid machinery commonly found in engineering practice. Studying and improving the centrifugal fans’ operating efficiency and overall performance in actual operation is of great significance for improving the EU of fluid machinery and implementing energy conservation and emission reduction (ECER) policies [5]. In actual operations, the working conditions of wind turbines in the power industry often work under harsh conditions. It is urgent to monitor the state of wind turbines effectively and ensure the stability, safety, and economic operation of wind turbines exposed to sudden changes in wind speed, sand dust, rain, snow, humidity, typhoon, lightning, and other harsh atmospheres for a long time. Thus, building a visualization system for the remote centralized control center of fluid machinery can facilitate remote monitoring and automatic operation management. It helps achieve maintenance, operation, and logistics management, directing the future development trend of smart machinery. In addition, it is a way to maximize the comprehensive utilization benefits of wind farms [6].

The concept of digital twins was first proposed by Dr. Michel Grieves, a professor at the University of Michigan in the United States, in 2003. Originally, it was positioned as “a set of virtual information structures that comprehensively describe potential production or actual manufacturing products from the micro atomic level to the macro geometric level” [7]. Later, some scholars described it as “the digital representation of real-world entities or systems”. In 2012, Glaessegen et al. gave the widely recognized definition of digital twins—that it is a complex product simulation model integrated with multi-physics, multi-scale, and probability, which can reflect the status of real products in real time [8]. Until 11 November 2020, the New Generation Information Technology Industry Standardization Forum, sponsored by the China Electronics Technology Standardization Institute under the Ministry of Industry and Information Technology, released the “2020 Digital Twins White Paper” [9] led by the Ministry of Industry and Information Technology. As an important research achievement under the background of new infrastructure construction, the white paper analyzes the current technological hotspots, application fields, industrial conditions, and standardization of digital twins in China. The appearance of the white paper reflects the increasingly profound impact of digital twins on China’s economic and social development. It can be seen that modeling, simulation, data fusion, and other technologies promote digital twins [10]. With the emergence of digital twin technology (DTT), people are realizing the value of information and numbers and foresee the correctness of physical entities. Thereby, they can avoid unnecessary risks and high-cost waste in the real world. In essence, DTT is a rehearsal in which digital information replaces physical entities so that digital value can be truly reflected.

DTT and the Metaverse [11] can provide more intelligent solutions for managing, monitoring, and maintenance of different parts of fluid machinery, such as fans, pumps, and bearings [12]. Following edge video intelligent platforms, distributed machine learning platforms, and other technologies, the video perception and intelligent analysis systems are sure to be a future development direction to conduct multi-dimensional perception and analysis of equipment and the environment. That is the new-generation, DTT-based intelligent machinery management model. Modern fluid machinery has seen extensive applications in the electric power industry, water conservancy projects, chemical industry, and petroleum industry. More importantly, it occupies a leading role in water conservancy projects as an energy transmission function. In particular, pump-type fluid machinery can transform the mechanical energy on the prime mover into liquid energy [13]. Nowadays, ECER has been the pursuit of the numerical simulation and testing of the flow in the pump, the research of pump theory and design methods, and the research direction of new technology [14]. DTT and the Metaverse present a multi-domain system, including fluid, electromechanical, electromagnetic, and thermal aspects [15]. At the same time, they can also offer user-friendly man–machine interfaces (MMIs) to monitor pressure and flow measurement values in the design and application of fluid machinery pumps and infer the real-time running speed and hydraulic state of the pumping system [16].

Therefore, this review collects, analyzes, and classifies the application status of pumps and fans in fluid machinery through literature research and comparative analysis from the perspective of DTT and the Metaverse. The relevant literature in the past 5 years has been selected for classification, summary, and analysis accordingly. Next, the research results are discussed. Lastly, future research on fluid machinery is prospected, aiming at the existing problems of fluid machinery in the simulation model, maintainability test, visual monitoring, and other aspects.

## 2. Methods

### 2.1. Literature Review

By reviewing and classifying the literature of DTT and the Metaverse in fluid-machinery-oriented applications in Google’s academic field, this review investigates the current situation of pump-type and fan-type fluid machinery in the past 5 years (2018–2022). Several keywords were chosen, including “digital twins + fluid machinery”, “digital twins + pump”, “digital twins + fan”, “Metaverse + pump”, and “Metaverse + fluid machinery”. Based on these keywords, the relevant technical research and algorithm (model) literature was searched. Following a check on literature titles, abstracts, and keywords, over 100 articles were selected and downloaded for literature review and specific analysis. The purpose was to investigate the specific changes and status quo brought about by different application fields of fluid machinery supported by the new intelligent technologies: the Metaverse and DTT. We look forward to the future orientation for existing problems to provide a reference for scholars and students studying fluid machinery.

### 2.2. Digital Twins and the Metaverse

#### 2.2.1. Digital Twins

Digital twins aim to build a “complete and independent mirror” of the physical world in the digital world through digital means. The mirror model can maintain the real-time interactive connection with physical entities and realize the understanding, analysis, and optimization of physical entities through simulation, verification, prediction, and control of the whole life cycle process of physical entities through historical data, real-time data, and the algorithm model [17]. In simpler terms, digital twins clone a device or system as a digital version [18]. The most dominating feature of this clone is the dynamic simulation of solid objects. In other words, digital twins are a dynamic process [19]. DTT relies on digital visual modeling and industrial simulation technology for implementation. Digital models help drive physical entities, and in turn, physical entities drive digital models, realizing the virtual reality fusion bidirectional mapping model in a real sense. The dynamic process of digital twins is illustrated in Figure 1.

As shown in, the collection and application of twin data realize equipment interconnection. Based on this, servers can display the equipment, equipment regulation, and regulation reliability data on mobile phones, pads, and other terminal devices. Next, users can understand and give feedback on the actual situation at any time. Digital twins mainly cover modeling, simulation, and data fusion technologies. Based on modeling and simulation technology, more data sets are added for data fusion calculation. This is a simple flow of digital twins from the basis to the core.

The first application of DTT can track back to the aerospace field with high technical content of large and complex mechanical equipment, such as aircraft and rockets, featuring zero error tolerance [20]. The US Air Force Research Laboratory used DTT to maintain the delicate fighter airframe. Later, General Motors (GE), a giant in the industrial manufacturing field, also pivoted its attention to DTT. Soon, with the attention of Siemens, Germany, DTT began to sweep the internet and the industry until today [21]. In industrial manufacturing, modifying the design and assembling product components usually demand repeated attempts and consume extensive human and material resources. DTT can establish a virtual space for industrial production to help engineering designers observe the external changes in products and dig into the dynamics of internal parts. Thus, DTT was gradually applied in manufacturing, industry, cities, battlefields, and other scenes, along with the maturing simulation technology. Figure 2 exemplifies the simulation applications of DTT in four scenarios.

#### 2.2.2. The Metaverse

Initially, the Metaverse was a concept coined in the science fiction *Snow Crash* by American writer Neil Stephenson in 1992 [22]. Facebook, the world’s largest social media platform, renamed itself Meta on 28 October 2021, triggering the global Metaverse research and development (R&D) boom and technological competition [23]. Internet tycoons, including Microsoft, Google, and Apple, have started their journey exploring the Metaverse. In addition, experts and scholars from different fields are trying to study the Metaverse from different perspectives. To quote Rabindra Ratan, associate professor of Media and Information Science at Michigan State University, “The Metaverse has three key features: realism, interoperability, and standardization”. The Metaverse integrates various sciences and technologies, and this mixture of science and technology superimposed in different directions is, in essence, science and technology. These different sciences and technologies have distinct goals until they are integrated to achieve a common goal—building the Metaverse. These technological bases of an integral Metaverse can be singled out, including virtual reality (VR), augmented reality (AR), DTT, internet of things (IoT), cloud computing (CC), artificial intelligence (AI), and fifth-generation mobile communications (5G). VR can project the real world into the virtual world compared to AR, which hopes to reverse this process [24]. In contrast, DTT strives to clone entire physical entities in the real world with digital mirrors. The IoT realizes a close connection between people and people, people and things, and things and things. CC attempts to integrate distributed computing power worldwide. AI aims to empower the learning ability and thinking mode of the machine. Lastly, 5G is committed to realizing timely, efficient, high-capacity information transmission. By integrating these technologies, a simple prototype of the Metaverse has been formed. Figure 3 expounds on the technical bases of implementing the Metaverse.

The Metaverse can be understood from the following technical features:

Full-Body Immersion Experience: Extended Reality (XR)

XR combines AR [25], VR [26], and mixed reality (MR) [27] to integrate virtual content with real scenes. The digital (virtual) space is generated through head-worn devices. The equipment’s key processing unit decorates the virtual scene where a full-body immersive sensory experience becomes realistic. Users can almost feel like they are roaming the actual space while being immersed in the virtual environment.

Efficient and Convenient Operation: Human–Computer Interaction (HCI) [28]

The virtual ⇌ reality interaction logic in the Metaverse: People’s behavior and consciousness are induced by computers, the results of which act on the twin entities. Finally, people’s consciousness or instructions are outputted to external AI devices. The interaction logic is a fundamental technical basis of the Metaverse connecting the physical and virtual worlds, namely an HCI process. High-end input technologies include computer AI sensors and brain–computer interfaces (BCIs). Currently, BCI technology is still in its infancy.

Digital Mapping Universe: DTT

The concept and application of DTT have been described in detail in the previous section. Real objects’ native data can be recorded in real time using DTT, then “copied” to the Metaverse, and displayed through 3D modeling technology. The real-timeliness of the digital twins’ data ensures that the image parameters of virtual objects in the Metaverse can also faithfully reflect their real conditions.

Functions of Free Editing: 3D Technology and AI

The “parallel universe” the Metaverse has presented is pluralistic and is the mapping of the real world but does not need all elements to be a twin of the world. The Metaverse tremendously releases people’s subjective initiative and creativity. People can edit and create freely, set different environmental parameters, simulate different life scenes, reconstruct different virtual things, and verify hypotheses.

Safe and Reliable Transaction: Blockchain Technology (BCT)

Transactions are involved throughout the creation, social interaction, shopping, or gaming in the Metaverse. Uniqueness and confidentiality features ensure the security of digital assets in BCT. The edited and created content of the Metaverse falls into the user’s personal digital assets, enriching the Metaverse information. BCT guarantees security for these digital assets. It promotes the decentralization of the Metaverse creation and the Metaverse economic model’s openness, flatness, and equality.

### 2.3. Researching the Fluid Machinery Pump from the Perspective of Digital Twins and the Metaverse

The Metaverse connects the real and virtual worlds and is the carrier of human digital survival and migration. Due to the increasing complexity of fluid mechanical systems, mechatronics and intelligence have become the development trend and the required diagnostic systems are also gradually becoming intelligent. There is an urgent need for intelligent fault diagnosis technology that integrates signal analysis, modeling, and knowledge processing [29]. Digital twins and Metaverse technologies provide new solutions for the intelligence of mechanical systems. As the technology of the Metaverse matures, relevant application scenarios will be widely used in various fields, including mechanical pumps. Exploring the modeling and simulation method of integrating equipment digitalization into digital twin and Metaverse scenes is of great significance. The object of fluid machinery is mechanical equipment, which uses a fluid (liquid or gas) as the working medium and energy carrier [30]. One of the fluid machineries is fan-type fluid machinery. Fluid machinery can be modulated into two categories: a liquid-related module (or hydraulic machinery) and a gas-related module (thermal machinery) [31]. The internal flow of fluid machinery is tricky and is characterized by several features:Three-dimensional flow parameters (e.g., velocity, pressure, and temperature) are all functions of three spatial coordinates.Viscosity: The actual fluid is always viscous, and the viscous flow inside fluid machinery is generally turbulent.Unsteady: The flow parameters will change with time, and the space–time scale will span a large range, sometimes 5–6 orders of magnitude.Compressibility is generally considered when the Mach number of gas flow is higher than 0.3.

Additionally, according to the application background, there may be gas–solid two-phase flow (dust inclusion) and gas–liquid two-phase flow (water pump cavitation) inside the fluid machinery.

Notably, the pump occupies a large proportion of the fluid machinery. It is a general-purpose machine for conveying fluid and increasing fluid pressure [32]. Pumps can work wherever there is fluid delivery, lending to water conservancy, petroleum, and chemical industries. They are also found in urban water supply and drainage, metallurgy, transportation, and other industrial sectors. According to the principle of action, pumps can be divided into three categories:Positive displacement pump: The periodic change of volume transports and increases the fluid pressure, including piston, plunger, diaphragm, and gear pumps.Vane pump. The high-speed rotating impeller inside transfers energy to the liquid, increases the pressure, and transports fluid, including centrifugal, mixed-flow, and axial-flow pumps.Other types of pumps. Hydrodynamic pumps, such as jet pumps and water hammer pumps, use hydrostatic pressure or kinetic energy of fluids to transport liquids.

#### 2.3.1. Research Achievements of Fluid Machinery Pumps from the Perspective of Digital Twins

Indeed, digital twins have proved their technical and intelligent value in architecture, engineering, construction, and facility management. Thus, the DTT-based asset monitoring and anomaly detection system and its data integration method based on the basic class of the extended industry have gradually been applied in daily operation, maintenance, and management. For example, in an urban water supply system, assets operate under varying loads determined by human demand. Lu et al. argued that the DTT-based new system could detect the pump anomaly continuously. The proposed system helped detect the anomaly of the pump efficiently and effectively and also helped in automatic asset monitoring in the pump operation and maintenance [33].

Chen et al. (2022) studied the heating, ventilation, and air-conditioning (HVAC) system by modeling its complete life cycle using the digital twin framework. The rectified linear unit (ReLU) was chosen to activation the network model. As a result, the mean absolute error (MAE), root-mean-square error (RMSE), and determination coefficient (R^2^) of nine multi-BLS chillers reached 9.04, 15.20, and 0.98, respectively. The proposed multi-BLS model outperformed several traditional models in prediction accuracy and response time [34]. Pedersen et al. (2021) modeled the urban water supply system through DTT. The observed changes in physical twins (with urban growth) were coupled with the simulation model and provided a similar conclusion [35]. Based on these analysis, the DTT-based application cases in equipment fault diagnosis of a centrifugal pump unit is explained in Figure 4.

As per, the geometric model of the twin unit describes the geometric parameters and relationships of the entities of the centrifugal pump unit. Mainly, the geometric parameters include the shape, position, size, and tolerance of the centrifugal pump, motor, coupling, and base. The relationships reflect the assembly relationship between the centrifugal pump, motor, and coupling and the assembly relationship of the parts in each component. The model can analyze failure, performance, optimization, and other dimensions. Each dimension analyzes the influencing factors of health status from different perspectives to form a multi-dimensional analysis method by combining data analysis and model analysis. Finally, multi-dimensional and multi-mode stereoscopic analysis is realized. The evolutionary model is a virtual model of physical entities at different times in the iterative process when the running time increases and the surrounding environment changes. An evolutionary model can help users grasp the health status of the equipment promptly.

In the manufacturing system, technical building services, such as cooling towers, are the driving factors of resource demand, and they also fulfill the important mission of keeping production running. Blume et al. (2020) suggested data-driven DTT models for technical building service operations. A case study was completed on the industrial building cooling tower in Germany to improve system understanding and performance prediction for successful operation management [36]. Moreover, as the core data service infrastructure, data centers in the manufacturing system have surged, leading to increased EC, hindering China’s sustainable development goal of ECER. In addition, data types of cooling towers were small in number and poor in quality. He et al. (2022) designed a convenient, rule-based data-preprocessing framework based on physical laws against the strong coupling of multiple variants. These engineering data were preprocessed as more reliable information to improve performance or train-specific models. Consequently, the designed framework adapted well for preprocessing multi-variant engineering data [37].

The physical ⇌ virtual entity transformation can also be realized using digital information in intelligent manufacturing. Fusing DTT and AI can explore the temperature rise boundary in digital data communication and excavate more data pairs. It is also conducive to the AI models’ robustness and energy efficiency. For example, Zhang et al. (2022) introduced a DTT-based collaborative framework for complex product design, manufacturing, and service integration. The model was used to analyze process integration, data flow, modeling and simulation, and information fusion. The core features and technologies of service-oriented manufacturing, service, design, and manufacturing monitoring were discussed. Finally, the feasibility of the proposed DTT-based framework was verified by a manufacturing case of a self-balancing multistage pump [38]. Thus, the emergence of digital twins provided an opportunity to accelerate the integration of complex product design, manufacturing, and service. To illustrate another case, in the machine tool sub-unit, Bernini et al. (2022) developed a DTT-based prediction method and applied it to the machine tool hydraulic unit. Three components were considered: pump, sensor, and valve. The DTT system simulated healthy and faulty machines, saving time, overhead, and expensive experimental activities [39].

Furthermore, when the measured fault states in intelligent manufacturing are insufficient, DTT allows fault diagnosis. In this regard, Xia et al. (2021) explored a DTT-based mechanical intelligent fault diagnosis (IFD) framework by integrating deep transfer learning (DTL). Specifically, the machinery’s twin model was built based on the simulation model by updating the continuous measurement data from physical assets. Finally, the new method was tested in a case study of the fault diagnosis of the triplex pump. The experimental results showed that the new method achieved an excellent IFD effect under a limited amount of measured data and beat other data-driven approaches. The finding proved that DTT has become the key technology of intelligent manufacturing [40].

Throughout the long-term experience, a powerful reference has been formed for the oil and gas (O&G) pipeline industry. It lacks specific standards and technical and economic frameworks to mitigate the obstacles the CO2 transport system faces. Thus, it is imperative to study the impact of impurities and elevation changes on a pipeline pressure drop to ensure continuous flow through the precise positioning of the pump station. In this respect, Sleiti et al. (2022) proposed a DTT-based integrated five-component frame for CO_2_ transportation through pipelines [41]. This warranted another way to the system parameterization of digital twins during O&G pipeline transportation. However, data-driven algorithms still needed to be developed to predict the system’s dynamic behavior. Meanwhile, in the O&G industry, the digital twins were supported by the digital mainline connecting the entire product life cycle data. Thus, processes, assets, and projects could be mirrored or simulated in real time to generate valuable insights. Nevertheless, the realized value was often challenging to quantify and link to the DTT in actual operation.

Similarly, Pashali et al. (2021) tried to remotely diagnose complications during pump operation, lifting leaks, and the operability of measurement systems. An intelligent algorithm was used based on expert rules, DTT, and machine learning. It was revealed that DTT helped predict the operating state of the oil well and the target parameters. As a result, additional operational costs for stabilizing other oil well production after the end of the main process were reduced [42]. Oil well construction is a complex multi-step process that requires decision making at each step [43]. Surely, there are methods and tools for monitoring oil well construction operations, but there is still no method or tool that can evaluate potential action sequences and scenarios and select the best action sequence. Accordingly, Saini et al. (2022) defined a DTT-based general iterative method to select the optimal action sequence [44].

Clearly, pumps are commonly used in industrial and residential applications, from water supply systems to O&G processing plants. These rotary hydraulic presses have a great impact on the EC of the global industry. This is because of their huge number and because of their continuous operation. Digital twins can potentially improve the prediction and mitigation of pump facility failures, thereby improving overall availability and the financial outlook of the project [45]. The stricter preciseness and supervision provided by the DTT-based platform can better remotely monitor and control O&G assets. In addition, abundant experimental data-driven models have been successfully deployed, and the predictive maintenance field will benefit from more industrial case studies. The investigation of model scale capability will benefit the industry that is studying large-scale implementation.

#### 2.3.2. Application and Research Results of the Metaverse in Fluid Machinery

The migration from a set of independent virtual worlds to an integrated network of 3D virtual worlds or the Metaverse depends on progress in four areas: immersive realism, ubiquitous access and identity, interoperability, and extensibility [46]. Vector field visualization at local critical points is a challenging task. Regarding this, Yang et al. (2022) proposed a learning-based multi-task super-resolution method to refine vector fields and enhance the visualization effect, especially in critical areas [47]. The novel approach was verified as an efficient end-to-end architecture suitable for the training and reasoning stages. It simplified the key vector field visualization pipeline and improved the visualization effect. The immersion of the Metaverse was realized through XR technology. Vortex methods have been developed and applied to analyze complex, unsteady, and eddy currents related to problems in various industries. They feature simple algorithms based on flow physics. For instance, Mahmood et al. (2021) explained the attractive application of the advanced vortex element method and its contribution to future-generation computational fluid dynamics (CFD). Along with CFD, these methods were introduced into the epoch-making application of simulating the unsteady flow around the open channel and the virtual operation of fluid machinery (pumps and fans) [48].

The signal characteristics of a multi-wideband in fluid machinery hindered the popularization of some narrowband demodulation methods, such as Kurtogram [49]. Such being the case, Wu et al. (2021) fused a multiple demodulation band selection, Enkurgram [50], by combining the energy factor and the shape factor. The fusion method presented excellent demodulation ability in simulation analysis and fluid machinery applications. Lastly, the effectiveness of Enkurgram was verified and provided with different noise interference conditions by simulation signals. The interference chose white Gaussian noise, random pulse interference, periodic pulse interference, and the pollution-modulated wave. The result shows that the proposed method effectively processed the actual signals of the centrifugal pump and propeller.

The Metaverse is undoubtedly a viable solution for fault detection of fluid machinery. Mining the fluid machinery’s prior fault conditions was a research difficulty. Against this dilemma, Arpaia et al. (2020) discovered a hidden Markov model (HMM) method by only training the normal machine operational data for fault detection [51]. Successfully, the trained model was combined with the goodness-of-fit (GoF) test to detect the error status by processing online data. Next, it was verified in the low-temperature cooling test of the screw compressor. Likewise, based on the automobile electronic pump, Si et al. (2020) proposed a CFD-oriented multi-objective design process and an intelligent optimization method to improve energy transfer efficiency. By understanding the flow loss mechanism in the pump, a numerical prediction model for pump performance was established [52]. It was observed that the maximum efficiency of the pump increased by 4.2%, so the proposed method was effective.

In addition, regarding the preventive maintenance of hydraulic pumps with external gears, Ritucci et al. (2021) used AR technology to step by step instruct operators to comprehend the dynamic disassembly sequence, in the form of animation, by programming applications for handheld Android devices (mobile phones) [53]. By increasing virtual buttons on the AR platforms, operators could start and pause/resume animations at any moment to observe the sequence creation steps with refined detail. Thus, it was convenient for device maintenance.

The pump network (including main and auxiliary pumps) supplies power for the circulating water system (CWS), while the turbine network recovers excess energy. Gao and Feng (2017) mathematically modeled the power target in the fluid machinery network. The model’s applicability was verified through examples [54]. The CWS was commonly used for cooling in the process industry, and its EC greatly impacted the energy performance of the whole plant. Gunder et al. (2018) designed a Metaverse-based observer or the reciprocating positive displacement pump’s valve. The observer could reconstruct the entire flow field in the valve with only two measurements. The increased observability of the reduced order model could be used to select the appropriate measurement location [55]. Next, Feng et al. (2020) used the alternating shortened Rayleigh—Plesset equation (RPE). They aimed to improve the predictability of the traditional cavitation model’s cavitation flow. Simultaneously, the homogeneous flow hypothesis was chosen to design an alternative numerical model. They used the novel transmission-based model in the numerical research of the turbulent cavitation flow in the axial flow pump. The finding corroborated that the cavitation performance predicted by the proposed model fitted better with the experiment than the Schnerr–Sauer model [56].

The renormalization group k-∈ turbulence model and the fluid multiphase volume method were used for the liquefaction pump. The leakage flow in the axial clearance was modeled for the 2BEA-203 liquid ring pump through numerical simulation. The numerical results showed that the axial leakage flow reduced the inlet vacuum and efficiency of the liquid ring pump, and the gas–liquid two-phase flow in the axial clearance area was separated entirely. The research found that these eddies were mostly formed in the suction and compression areas of the clearance area and significantly affected the pump’s overall efficiency. By comparison, a self-priming pump is an important fluid machinery widely used in disaster relief and emergency areas. The impeller is the only rotating unit of the self-priming pump, playing a vital role in the power capacity of the pump. Chang et al. (2021) proposed impellers with different hub radii. The relationship between the transient characteristics and the hub radius was obtained by comparing each scheme’s internal flow characteristics, blade surface load, pressure fluctuation characteristics, and radial force distribution. Relevant hydraulic experiments were carried out. The difference between the experiment and the calculation was no more than 3%, ensuring calculation accuracy [57].

The previous subsections have introduced different types of pumps. In particular, centrifugal pumps generate strong noise during operation, damaging the pump unit and personnel. Thus, noise reduction in centrifugal pumps is the research focus. Concerning noise reduction, Dai et al. (2021) researched centrifugal pumps with a specific speed of 117.3. The effect of different bionic structures on the noise reduction performance of centrifugal pumps was investigated through the bionic transformation of centrifugal pump blades [58]. The test was carried out on a closed pump test platform, including external characteristics and a flow noise test system. The results showed that the pit structure had little influence on the external characteristic parameters, while the sawtooth structure had a greater influence.

The booster pump system can control the number of revolutions through the inverter by connecting two or more vertical or horizontal centrifugal pumps in series. High efficiency and energy saving are the most attractive features of the booster pump system. These aspects can be improved by measuring the flow of each pump to control the operating conditions of a single pump. To improve operation, booster pump systems with flow sensors and control algorithms are critical. Therefore, Rakibuzzaman et al. (2022) developed a turbine-type flow sensor. Afterward, the flow sensor was optimized using CFD theories and verified through experiments [59].

In the O&G industry, the oil pump maintenance task is exceedingly complex, which is highly urgent for the economic operations of industrial enterprises. One reason is that pumping equipment is widely distributed in various fields of O&G enterprises. AR technology in the Metaverse can reduce equipment maintenance time by reducing the time to search and process maintenance and feedback information.

In addition, the Metaverse benefits businesses and industrial fields and helps present more vivid teaching experiments. Kapilan et al. (2021) trained mechanical engineering majors in the virtual laboratory of fluid mechanics. It was observed that at least 90% of the participants were satisfied with the virtual laboratory. The respondents recounted that their learning process had been improved through the virtual laboratory experiment. Hence, using the virtual environment for experimental teaching was of great help to the overall teaching effect [60]. Koteleva et al. (2021) studied the effectiveness of the collaboration between AR and dynamic simulation systems in O&G pump maintenance. The system structure and the development of the AR-based system application were designed and tested with Microsoft HoloLens V2 [61]. It was revealed that the AR + dynamic simulation system optimized the O&G pump’s overall maintenance efficiency. Nevertheless, the timeliness and authenticity of the virtual simulation system warranted further verification.

Apparently, fluid machinery’s application backgrounds are diverse, involving different sizes of research objects. For illustration, the diameter of the industrial ventilation compressor may be more than 700 mm compared to the household automotive turbocharger with about a 45 mm diameter. The construction of the test bed varies with different application backgrounds. CFD theory is frequently used to numerically solve the control equations of the internal flow of fluid machinery to obtain internal flow field information. Alternatively, an experimental study may be used, namely internal flow testing of fluid machinery. Under experimental conditions, pressure and temperature sensors, a laser Doppler velocimeter, an image particle velocimeter, and other fluid-testing methods can measure the internal flow parameters to obtain internal flow field information [62]. Digital twins and Metaverse technologies shed new light on the research of fluid machinery. The twin model of pumps and fans in fluid machinery in different fields through the one-to-one reduction of the real environment to the dynamic virtual environment can help operators achieve remote control. The simulation test environment built by the virtual environment greatly saves costs.

### 2.4. Research of Fans in Fluid Machinery from the Perspective of Digital Twins and the Metaverse

There are three common types of fans:(1)Centrifugal fan

Centrifugal fans [63] rely on the input of mechanical energy to increase gas pressure and discharge gas machinery, and they are a driven fluid machinery [64]. Nevertheless, the actual operation often deviates from the design conditions, resulting in a decline in operating efficiency, seriously affecting energy efficiency. Centrifugal fans are widely used in factories, mines, tunnels, cooling towers, vehicles, ship and building ventilation, dust and cooling, ventilation (air induction) of boilers and industrial furnaces, cooling and ventilation in air-conditioning equipment and household appliances, grain drying and selection, and wind tunnel wind source and hovercraft inflation and propulsion.

(2)Axial-flow fan

In an axial-flow fan, the air flows in the same direction as the axis of the fan blade, including the electric fan and the external fan of the air conditioner [65]. The name “axial flow” comes after the fact that gas flows parallel to the fan axis. Axial-flow fans are usually used when flow requirements are high and pressure requirements are low [66]. The axial-flow fan is fixed in position and moves the air. Small axial-flow fans feature low power consumption, fast heat dissipation, low noise, energy conservation, and environmental protection, with a wide application. A large axial-flow fan has a simple structure and is stable and reliable, featuring low noise, a large air volume, and wide function selection.

(3)Diagonal-flow fan/mixed-flow fan

The diagonal-flow fan/mixed-flow fan [67] is a fan between the axial-flow fan and the centrifugal fan. The impeller of the diagonal-flow fan makes the air move both centrifugally and axially. The movement of the air in the shell is a mixture of axial flow and centrifugal motion. It combines the characteristics of axial and centrifugal-flow fans. However, diagonal/mixed-flow fans look more like traditional axial-flow fans [68], with the curved plate blade welded on the conical steel hub. The flow is varied by changing the angle of the blades in the inlet housing upstream of the impeller. The enclosure may have an open inlet, but more commonly, it has a right-angled bend shape that allows the motor to be placed outside the pipe. The discharge shell expands slowly to slow air or gas flow and convert kinetic energy into beneficial static pressure. It is mainly used for mine or tunnel ventilation.

#### 2.4.1. Research Achievements of Fans from the Perspective of Digital Twins

Over the years, the hydrodynamic design of fluid machinery has relied heavily on empiricism and experimental observation, as has the research on fans [69]. Centrifugal impeller manufacturing is moving toward a new paradigm, intending to improve efficiency and competitiveness through Industry 4.0 and intelligent manufacturing [70]. Making centrifugal impellers developable and regular has become a key technology to significantly improve machining efficiency and save costs, although it may have a corresponding negative impact on aerodynamic performance. Zhou et al. (2021) proposed a DTT-based optimization strategy by simultaneously considering machining efficiency and aerodynamic performance. They built a specific 5D digital twin model to simulate and analyze the harmful effects of aerodynamic performance and the internal flow field. It was proved that DTT could combine the real and virtual worlds, allowing the geometric dynamic update and iterative optimization of the centrifugal impeller. Their findings were of great significance in effectively shortening the centrifugal impeller’s development cycle and saving costs [71]. Further, to redesign the impeller and blade, Anbalagan et al. (2021) verified the concept of digital twins in a virtual environment and discussed computer-aided design (CAD) modeling and manufacturing simulation methods. Specifically, DT and CAD were used to immediately design/redesign and manufacture the impeller and blade. Finally, DTT helped concurrently verify the design and manufacturing processes [72].

The IFD technique in the data twins system usually includes four parts: real-time prediction, result verification, digital twin model correction, and deep learning model reconstruction. The technical scheme of DTT-based IFD is shown in Figure 5.

Figure 5 shows that the vibration signal of the centrifugal pump unit is predicted in real time by using the data-driven fault diagnosis method. The diagnosis results are verified by the model-driven method, thus ensuring the reliability of the diagnosis results. The digital twin model is modified if the diagnosis result is accurate. Otherwise, the deep learning model is reconstructed, and the fault prediction is repeated until the diagnosis results are accurate. The self-correcting characteristic of the predictive maintenance scheme under the digital twin model ensures the synchronization of industrial equipment in the physical space and the twin model in the information space. At the same time, the closed-loop nature of digital twins is reflected in the visual model and internal mechanism that can describe physical entities. As such, the status data of physical entities can be monitored, analyzed, and reasoned. Thereby, process and operation parameters are optimized, realizing decision-making functions and improving the accuracy of judging the health status of equipment.

Given the nonlinear dynamics and uncertainties involved in the mechanical degradation process, the correct design and adaptability of the digital twin model is still a challenge. Wang et al. (2019) designed a DTT-based reference model for rotating machinery-oriented IFD. They discussed the requirements for building a digital twin model by suggesting a model-updating scheme based on parameter sensitivity analysis to enhance its adaptability. The outcome showed that the digital twin rotor model constructed could accurately diagnose and was adaptive to degradation analysis [73]. Likewise, Ritto et al. (2021) built a DTT-based model for the damaged mechanical fan structure. A computational model based on discrete physical fields was used to study several damage scenarios. Therefore, these aforementioned selected applications could highlight every step of digital twin construction, including the possibility of integrating physics-based models with machine learning [74].

Underground heritage sites also need reliable and efficient ventilation systems because of constant humidity that can cause surface and structural damage [75]. To this end, a DTT-based dynamic ventilation system can help formulate a protection mechanism for underground sites and offset the adverse impact on physical equipment and system energy use. A web-based digital twin platform combined with IoT technology was established to achieve real-time control of the overall relative humidity level in underground heritage sites within the standard range. The suggested approach could be used in typical underground heritage sites. Meanwhile, the feasibility was verified for developing digital environments for real cases and developing ventilation systems to optimize their relative humidity schemes [76].

The power generation system’s nonlinearity in a power plant will affect the driver’s performance. Sometimes, more fans than required will be activated to ensure water temperature control, thus increasing EC [77]. Aiming at this problem, Alves de Araujo Junior et al. (2021) developed a DTT-based water-cooling system with auxiliary equipment. They determined the number of correct fans according to the operator’s decision [78]. In terms of wind power, with the progress of intelligent technology and accelerated wind power industrialization, China has formed an industrial basis for large-scale wind power development and application. Especially as big data technology (BDT), AI, and other digital technologies step into the wind power industry, intelligent wind turbines emerge as the times require. The wind turbine works following a simple rule: the wind drives the blades to rotate, and the blades drive the generator shaft to rotate for power generation. However, in wind power generation (WPG), the influence of the direction and speed of natural wind can be tricky.

Simply put, a low wind speed will fail to turn the blades, while a strong wind might damage the internal parts of the blade or even the blade itself. Hence, a wind vane and an anemometer are designed on the fan. According to the wind direction and speed, the wind energy can be fully used by turning the head and changing the angle of the fan blade. Only partial results and parameters are fed back to the sensors. It is still impossible for managers to accurately understand the operation status of the key parts of the wind turbine or to actively adjust them under special circumstances [79]. In these circumstances, DTT can help establish twin fans to conduct all-round monitoring of the fan [80]. The twin fan can comprehensively use sensing, computing, modeling, deep learning, and other technologies. It combines physical, mechanism, and other digital approaches to model the fan. Thereby, it covers description, diagnosis, prediction, and auxiliary decision-making functions to realize the digital twin display of the fan [81]. Many experts and scholars have given their responses with regard to DTT applications in wind power.

Wind turbines are one of the main renewable energy sources, bringing about sustainable and efficient energy solutions [82]. They release zero carbon emissions that pollute the earth. Due to the unpredictability of wind speed, monitoring wind farms and predicting WPG are complex problems [83]. This is reflected in the following aspects:Production monitoring

Sahal et al. (2021) proposed a DTT collaboration conceptual framework to intelligently detect unstable operational data through operational data in manufacturing systems. The framework offered an interaction mechanism for understanding DTT status, interacting with other DTT-based models, and sharing common semantic knowledge. It described the energy-4.0-based IFD cases of wind turbines to test the proposed framework and DTT collaboration on identifying and diagnosing potential failures [84]. De Kooning et al. (2021) devised a DTT-based advanced structure and a virtual replica with the minimum computing load. A DTT-based wind energy conversion system was established, reaching an appropriate balance between model fidelity and computing load [85].

Fan blade monitoring

The blade is the “feeler” of the fan to feel the wind energy and is the first and heavier pressure-bearing component in several major fan systems. The intelligent monitoring system built using topology software can feed back the running status of blades in real time through the image pictures and results collected by oblique aerial photography. Oyekan et al. (2020) examined the creation of automated units for the maintenance, repair, and overhaul of wind turbine blades’ repair components. A DTT-based grinding process was developed and used to calculate the required grinding force parameters for effectively removing surface materials. The results ascertained that the proposed system could perform material removal, track the state of the fan blades during repair, and perform this operation within a closed-loop automated robot unit [86].

Online monitoring of fan operation

Yakhni et al. (2022) monitored the status of a ventilation system based on DTT. According to the free-body diagram and Newton’s second law, the equation of motion was obtained, and the specific frequency was adjusted in the twin to achieve the best simulation [87]. Chen et al. (2021) proposed a new concept of intelligent and semi-automatic man–network–physical systems to operate and maintain intelligent wind turbines. The proposal promoted human intelligence to the level of supervision. Particularly, high-level decisions made through the HCI would undermine autonomy, when needed. This research was undoubtedly a blessing for the increasingly complex operation and maintenance of wind turbines [88]. Additionally, with the rapid development of global wind energy, offshore wind turbines (OWTs) are increasingly used. The supporting structure of an OWT is easily damaged. The reliability of the structure warrants enhancement to prevent accidental failures and reduce operating costs through IFD, residual s life (RSL) prediction, and condition-based maintenance [89]. In similar scenarios, Fan et al. (2020) conducted a time-domain fatigue analysis of the multi-planar tubular joints of the OWT jacket substructure. The tubular joints were designed under local environmental conditions in Taiwan. The result showed that the fatigue damage caused by the power generation design scenario accounted for 90% of the total cumulative fatigue damage. Introducing DTT to analyze the fatigue damage in time would save costs substantially and reduce the material costs of OWTs [90]. Evidently, the application of DTT in wind turbines is relatively mature, especially in intelligent manufacturing and wind power research.

#### 2.4.2. Research Results of Fluid Machinery Fans from the Perspective of the Metaverse

The Metaverse is a virtual world that can be digitally interacted with the real world. In the real world, many scientific and technological innovation activities cannot predict the results due to complicated factors, or the prediction is astronomically costly. The Metaverse can serve as a panacea to offset these difficulties by accurately copying the real world, mainly relying on DTT. Especially, the distributed new energy industry involves multiple types of use (WPG, photovoltaic power generation (PVPG), and solar thermal power generation) and rich production forms. This has enriched the scenarios of energy use to a certain extent and increased unpredictability and risks. Thanks to the technical support of the Metaverse, energy production scenes with complex scenes, many variables, and more complex operations are completely copied into the virtual space to carry out virtual simulation tests in the complete life cycle: planning, design, construction, implementation, operation, maintenance, and adjustment. Metaverse-based simulation reduces the test cost in the real world and ensures the safety and reliability of distributed new energy use.

Galuppo et al. (2021) focused on the specific configuration of the system using direct condensation and dedicated fans in the field of waste heat recovery. The experimental results presented the models of all components in the system and the validation of evaporator and fan models. Special attention was paid to the effectiveness of fan speed and condensation pressure control to increase the net power output of direct condensation [91]. An axial-flow fan is a component of the blower used for ventilation in various industrial fields.

Pertaining to axial-flow fans, Li et al. (2020) studied the dynamic impact of configuration on blade performance and predicted the aerodynamic noise of variable-pitch axial-flow fans with guide vanes. The prediction using a large eddy simulation method showed that the Gurney flap improved the aerodynamic noise of fans [92]. Thus, although GF0.5-100 and GF2-50R are the preferred axial-flow fans, some necessary measures should be taken to reduce acoustic noise in practical applications. Targeting these issues, Kook and Cho (2021) used the maximum stress and safety factor. They verified the structural safety and weak area analysis results of the components under the rated operating speed of the axial-flow fan. The tip clearance reflected in the design was the rotation of the blade. To check whether there was interference with other components, the displacement was counted to verify the structural safety of the axial-flow fan [93].

In addition, for an axial-flow fan, there is a wide recirculation area downstream of the fan near the hub. To accurately predict the CFD parameters for such fans, Wang and Kruyt (2022) performed validation simulation on fans with small hub-to-tip diameter ratios by comparing experimental and calculated aerodynamic performance characteristics [94]. The two scholars adopted the CFD-based simulation strategy already verified on the baseline axial fan in 2022 and studied the impact of different eddy current distributions on the aerodynamic performance. The current CFD parameters showed that the design of free vortex distribution and polynomial vortex distribution met the expected pressure rise, and the overall static and overall efficiency significantly improved (maximum increase of 3.9% and 4.6%, respectively) [95]. Lee and Kil (2020) used the streamline curvature method to predict the flow and performance of axial-flow fans and made the empirical correlation between flow deviation and pressure loss. Their finding indicated that the actual (round) trailing edge is necessary. The fan’s main blade (no non-aerodynamic blade part) well represented the aerodynamic performance of the whole fan blade. It was recommended not to consider the suction head clearance [96].

Ye et al. (2022) used steady Reynolds averaged Navier Stokes and large eddy simulation models to conduct a numerical study. The aerodynamic performance and noise effects were studied for a single-stage variable-pitch axial-flow fan. It was concluded that the appropriately parameterized sawtooth trailing edge (STE) improved the span distribution of the vortex. In addition, the designed STE reduced the span size of the wake vortex and induced a more uniform distribution of the wake vortex. Thus, STE blades could provide a new method to control the aerodynamic noise of rotating fluid machinery [97].

Apart from axial-flow fans, cross-flow fans are used in several fields, such as air conditioning and aircraft propulsion, and have demonstrated their practicality in the ventilation system of hybrid electric vehicles. Himeur et al. (2022) used analytical modeling and experimental data to infer the performance of cross-flow fans in turbomachinery. Two different loss models were compared in detail to determine the performance–pressure curve of such fans. The internal flow field was explored through multidisciplinary research, CFD simulation, and optimization algorithms, and additional information about the eccentric vortex was obtained. The formula for the efficiency of the Eck/Laing cross-flow fan was obtained, and the experimental results were well verified, thereby achieving efficiency evaluation. Hence, the proposed method could effectively speed up the design cycle of cross-flow fans and predict their global performance [98].

Correspondingly, Han et al. (2022) studied the influence of three noise reduction methods on the rotor–stator interaction of diagonal-flow fans through numerical simulation based on CFD and Ffowcs Williams and Hawkins methods. There were many design variables in small diagonal-flow fans with rear guide vanes. The design parameters could interact with each other, resulting in performance degradation of the fan under actual operating conditions [99]. Zhou et al. (2022) combined Pearson correlation analysis (PCA) with CFD, the Latin hypercube experimental design, and the Kriging agent model. A method was advanced for designing multi-objective aerodynamic performance strength. The structural parameters of the rear guide vane of the diagonal-flow fan were optimized under uncertain aerodynamic performance. The flow field characteristics before and after the transformation were compared to select the optimal design model of the version. The test results showed that the total pressure of the optimized fan increased by 86 Pa and the noise decreased by 2.4 dB. Thus, the proposed method could optimize the performance of other types of fans [100]. This literature review shows that the Metaverse concept still lacks a unified measurement standard. Its application in fluid machinery is not mature enough, and more practical cases are lacking.

## 3. Research Summary of Digital Twins and the Metaverse in Fluid Machinery

The preceding section summarized the current state of DTT and Metaverse research in fluid machinery fan and pump devices. Indeed, digital twins and the Metaverse can perform real-time modeling, monitoring, analysis, prediction, control adjustment, and some degree of transformation on physical objects by combining machine learning, BDT, IoT, 5G, BCT, and other emerging technologies. DTT can assist in the development of industry knowledge, the analysis and prediction of the industry’s overall trend, and the generation of forward-looking recommendations. It has the potential to significantly improve complex problems in manufacturing, such as industrial chain coordination and comprehensive urban governance, as well as change the operating mode of various industries. The main achievements of the application of DTT and the Metaverse in fluid machinery are summarized in Table 1.

According to, an important component of fluid machinery, the pump, plays a role in applying DTT and Metaverse technologies in different types of pumps. However, there are differences in the authenticity and accuracy of the simulation and test results for low- and high-flow pumps. CFD is still the main technology and means of fluid machinery research. Only based on the DTT model and corresponding hydraulic technology can accurate prediction of pump failure state and energy efficiency management be truly achieved. The development of Metaverse technology is at an early stage, and the research literature on fluid machinery is not rich enough. It is limited to the advantages of VR and DTT. In other aspects, more cases are needed, and this phenomenon will gradually mature with Metaverse development. DTT is widely used in fluid machinery fans, involving different types of fans. For example, in power plants, DTT and Metaverse technologies can use a relatively small amount of data to create models with low false responses. It is easier to modernize older industrial plants because these plants usually do not have a large amount of stored production process data. The blade gear and test parameters of the low-pressure axial-flow fan with a small wheel tip diameter ratio can be changed to optimize its aerodynamic performance.

## 4. Discussion

For mechanical equipment, especially powered or rotating equipment, component resonance, imbalance, misalignment, lubricating oil pollution, gear mesh failure, motor blade failure, stator failure, rotor failure, and bearing failure are accompanied by changes in vibration information. The occurrence and development of equipment failures have their objective laws. Generally speaking, the performance or status of equipment in use gradually declines. When the fault reaches a certain degree, it leads to a sound change. During manual patrol inspection, sound is also used to identify whether there is a fault in the equipment. However, when the equipment is found to have a faulty sound, the equipment fault has reached an irreversible level. Based on the strong correlation between physical entities and virtual images, DTT brings a new perspective to solve problems for industrial scenes. By combining all kinds of explicit knowledge with tacit, structured, and unstructured knowledge, we can activate silent knowledge and data assets accumulated by the industry over many years, overlay real-time and quasi-real-time dynamic operation data records on traditional industrial models, build industrial digital twins, and help people re-understand and manage industrial manufacturing. It can be said that it is the industrial internet that has activated DTT.

Traditional industrial production has gradually changed to intelligent production. From the industrial transformation worldwide, new production forms, such as intelligent manufacturing, intelligent data, intelligent products, and intelligent services, have put forward new requirements and challenges to industrial enterprises’ production management and service mode. With the accelerated application of the new generation of information technology represented by cloud computing, BDT, and the IoT in industrial production, the interconnection between automated and intelligent products and production equipment has become an essential foundation for the construction and development of intelligent manufacturing. The industrial digital twin platform is an integrated platform that can realize the elastic expansion of the platform, equipment access, equipment operation and maintenance, and digital factory. Indeed, it builds the IoT infrastructure for the intelligent production and operation for enterprise customers. Digital, visual, and intelligent management and operation are then achieved. In contrast, equipment data run through the whole business chain: product innovation, production operation, and asset maintenance. These data promote industrial enterprises’ digital transformation and intelligent development.

In industrial development, due to the complexity of fluid flow phenomena occurring in fluid machinery, the overall performance of these machines cannot be fully predicted by analysis programs. Therefore, alternative modeling techniques, such as computer-aided numerical analysis, are usually used to predict pump performance. However, for various reasons, the actual performance of pumps and fans may deviate from their ideal behavior, which may change the performance curves provided by the manufacturer in the corresponding data sheets. DTT can bring the simulation environment closer to reality by replicating the behavior of the physical system with the help of experimental data.

For example, Chen et al. (2022), [101] explored the role of DTT + the Meta heuristic optimization algorithm in industrial manufacturing energy efficiency optimization. Based on the DTT-based machine tool model, the energy consumption in the milling process of computer numerical control (CNC) machine tools was studied. In addition, the particle swarm optimization (PSO) algorithm was introduced to optimize the milling parameters of the machining process. While ensuring processing quality, the proposed scheme improved processing efficiency and energy efficiency and reduced energy consumption. Thus, optimization algorithms can promote the industrial manufacturing process’s efficiency.

Multi-pump systems-oriented digital twins, for example, can be created in serial and parallel configurations. This can show how the pumps are linked in series or parallel to assist managers in adjusting equipment operating conditions. As a result, more efficient operating conditions can be achieved to respond to changes in the pipeline’s downstream conditions. However, the security problems of intelligent wireless terminals in digital twin biological networks cannot ensure wireless communication networks’ stable and efficient operation. This problem needs to be solved to ensure the normal operation of fluid machinery. In this regard, Feng et al. (2021), [102] proposed an interference source localization scheme based on a mobile tracker in IoT communication. The proposed model improved attribute-based encryption to meet the security and time overhead requirements of digital twin network (DTN) communication.

As a result, DTT-based applications in fluid machinery enable monitoring and predictive maintenance for internal structure operation. Analyzing the available research reveals that DTT and the Metaverse (immersion experience, HCT, and VR) provide a variety of solutions for studying complex fluid machinery systems. Many issues can be identified at the same time.

The development of a multiphase flow test bed for fluid machinery is primarily intended to address the dreadful operating conditions of hydraulic machinery (especially hydraulic turbines). Many rivers in China have high sediment concentrations. Sediment abrasion and cavitation erosion are common and serious problems, but construction is performed with inadequate domestic testing and design experience. As a result, fluid machinery has numerous flaws. Cavitation, blockage, blade wear or damage, overheating, surge, misalignment, leakage, and bearing damage are the most common. With the help of AI, fluid dynamics, sensors, IoT, and other technologies, DTT and VR technology in the Metaverse can help provide solutions for fault testing, early warning, daily visual monitoring of system operation, and other functions in the fluid machinery manufacturing system. Through machine simulation and virtual debugging, digital twins can reduce product risk, integrate different technologies, and collaborate with personnel via a unified data management platform. They can also maintain innovation and cost advantages by collecting and analyzing data from manufacturing workshops and machines. However, realizing digital twins and the Metaverse requires significant data and technical support. In addition, effective support of IoT devices is also required. Ashraf (2021), [103] argued that IoT devices intelligently collect data from each event and sent it for further processing. A necessary part of these devices is wireless sensors for building smart cities. Likewise, Cheng et al. (2022) [104] argued that with the continuous expansion of the IoT data center scale, the data center’s energy consumption is increasing and restricting the development of the data center. Therefore, in the specific application process, the enterprise should evaluate its technological preparations and capital investment that it could adopt reasonably.

Table 2 compares the advantages and disadvantages of these similar studies and the review reported here.

As can be seen in, similar conclusions are drawn by comparing the studies in this review with similar reviews of digital twins and the Metaverse in fluid mechanics and industry. For example, Gadekallu et al. (2022) [105] reported that the Metaverse aims to bring 3D immersion and personalization by using many related technologies to provide users with experiences. Although this proposal is great, a natural issue in the Metaverse is how to ensure the security of its users’ digital content and data. This is similar to the findings of this review, which demonstrate both the feasibility and limitations of the application of the Metaverse in different domains. Far and Rad (2022) [106] presented the definition, applications, and general challenges of digital twins and the Metaverse. The study then provides a three-tier architecture that links the physical world with the Metaverse through a user interface and also investigates the security and privacy challenges of using digital twins in the Metaverse. This is in line with the research logic of this review. Both show that the framework organization of this review is reasonable and prove the importance of using a blockchain in the Metaverse in the future for the development of the Metaverse in the industry.

It can be seen that DTT and Metaverse technologies are deeply combined with the equipment industry, with the advantages of digital fidelity, real-time interaction, foresight, and symbiosis. This integration plays an important enabling role in industrial transformation and upgrading. The industrial field has been applied from interactive digital twins to the coupling of different digital twins across traditional silos in utilities. In addition, the selection of different interpolation methods according to the magnitude of characteristic curve curvature can further improve the accuracy and automation of data processing through error correction of zero position and sensitivity, as well as nonlinear correction. With the development of science and technology, the application of DTT and the Metaverse in fluid machinery will appear in instruments and meters with high accuracy, which will improve the accuracy of the measurement and control system. At the same time, improvement of testing and simulation technology will reduce the errors generated in the testing process, which is also the direction of further research.

## 5. Metaverse and DTT Challenges, Potential Solutions, and Future Directions

Emerging technologies are advancing by leaps and bounds, accelerating the development and reform of various economic and social fields. In this context, DTT and the Metaverse, as key technologies and important tools to improve efficiency, can effectively play their roles in model design, data collection, analysis, prediction, and simulation. They help promote digital industrialization, industrial digitalization, and the integration and development of digital and real economies. The digital twin system plays a practical role in creating value for enterprises, transforming enterprises, and finding new value models for enterprises. Further, DTT can provide people with an authentic service experience. In turn, human beings will launch more diverse demands on the virtual world. Moral ethics will be challenged. Based on the interaction between the physical world and the virtual world, there is a possibility that virtual acts may indirectly damage the real world. The prevention and control of digital crimes and the legal construction of the virtual world will become new research topics.

DTT needs to perceive, diagnose, and predict the state of physical entities by using simulation, measurement, and data analysis to optimize physical entities and evolve their digital models. The data center is an important data transfer hub and business carrier platform infrastructure. However, some challenges have not been broken through in technology. Presently, DTT applications have been gradually launched in China’s intelligent manufacturing and engineering construction field. With the gradual development and maturity of the DTT system, there will be more industry scenarios to achieve cost reduction and efficiency increase by building digital twin applications. The following requirements and challenges need to be met:Digital twin representation is related to IoT equipment and must cover a wider range of abstract capabilities.Digital twin solutions must be allowed to model the relationship between IoT devices. Customers should be able to create new digital twin models specific to their verticals or use cases.The automation platform must provide a configuration infrastructure to support the creation of digital twins and associate them with different IoT devices.Creating automation scenarios on these “composite” objects should be seamless. That is, automation templates should be able to be assigned to these objects, not just to IoT devices. Any specific relationship between IoT devices should be resolved automatically during configuration.IoT devices should be able to send data at different times so that seamless streaming data merging is possible.Automation rules must be able to access configuration data, and these configurations will change over time. At run time, all automation rules must be aware of these changes.The IoT platform should allow the end user to create automation rules on the digital twin asset series, which should be completed during configuration. This will force different asset families to require different rules, which should be configured only once rather than on a single asset.

The Metaverse also faces challenges in different application scenarios. First, like the current internet, privacy is bound to be a big challenge. For example, when some services (e.g., virtual banks) are interwoven with reality, there may be a risk that virtual identity and real identity will be linked. Its dimensional universe is a huge opportunity for advertising marketing. In addition to virtual products in the VR space for users to appreciate, the Metaverse can also convey the stories of products or brands with a more interactive experience. However, there is bound to be a gap between virtual and real products, especially with regard to material, texture, smell, and other aspects. It is not easy to reproduce in the virtual world with current technology. Properly handling the disputes caused by this is a problem that needs to be considered. Finally, because the virtual reality experience is closer to the real world, compared with the computer screen, it is more likely to have an aversion to the excessive proximity of others. With the development of the Metaverse and technology, it can be predicted that more challenges will emerge, impacting the user’s usage habits.

## 6. Conclusions

This review selects pumps and fans in fluid machinery from the viewpoint of digital twin and Metaverse applications in fluid machinery. Next, a comparative analysis is conducted in the form of a table. In addition to relatively mature applications in smart manufacturing, DTT and the Metaverse play a vital part in developing new pump products and technologies. They are widely used in areas such as numerical simulation and fault detection in fluid mechanics for various pumps. However, there are relatively few application cases of the Metaverse. This is because the implementation of the Metaverse requires strong technical support (DTT, virtual reality, blockchain, etc.), and the current technical conditions are insufficient to fully support the full realization of the Metaverse, which is also reflected in this survey. A comparison with different similar studies reveals that the framework organization and overall findings of this review are of a practical reference value. However, there are also limitations. Because of the limited time, only part of the research literature from the database in the past 5 years has been surveyed. Follow-up work will conduct an in-depth investigation and analysis of other areas of fluid mechanics.

## Figures and Tables

**Figure 1 sensors-22-09294-f001:**
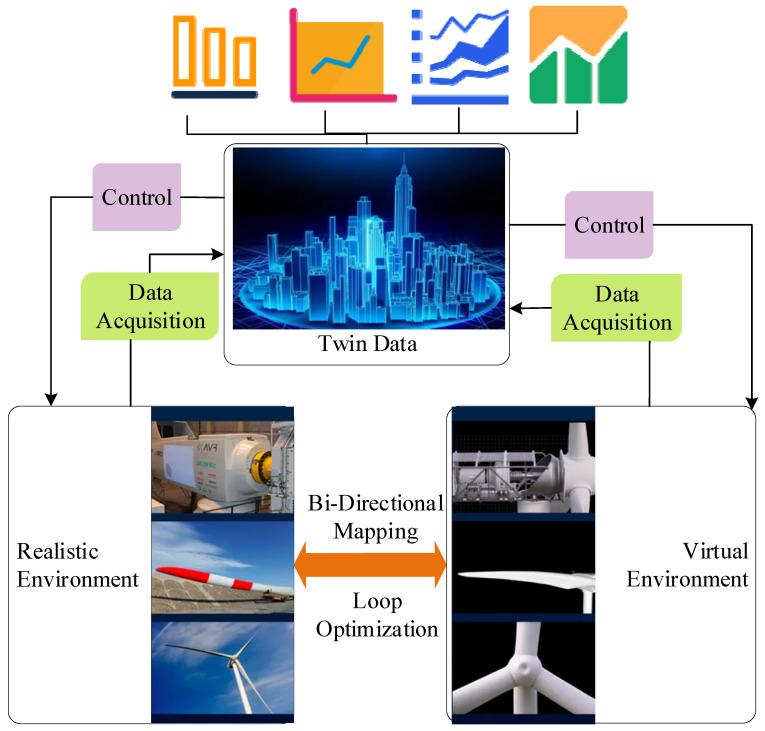
The digital twins technology (DTT).

**Figure 2 sensors-22-09294-f002:**
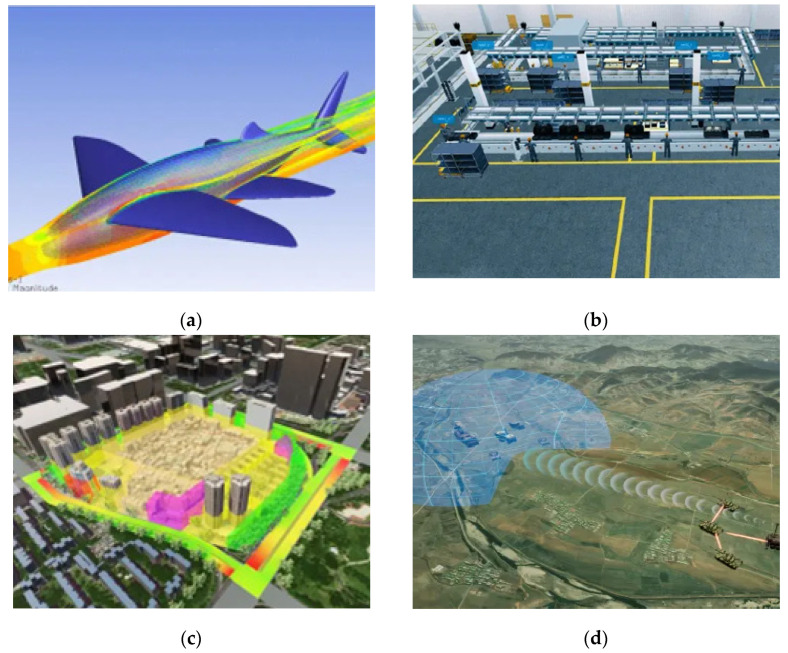
Examples of DTT simulation in different scenarios: (**a**) aircraft aerodynamic simulation, (**b**) factory simulation, (**c**) urban simulation, and (**d**) battlefield simulation.

**Figure 3 sensors-22-09294-f003:**
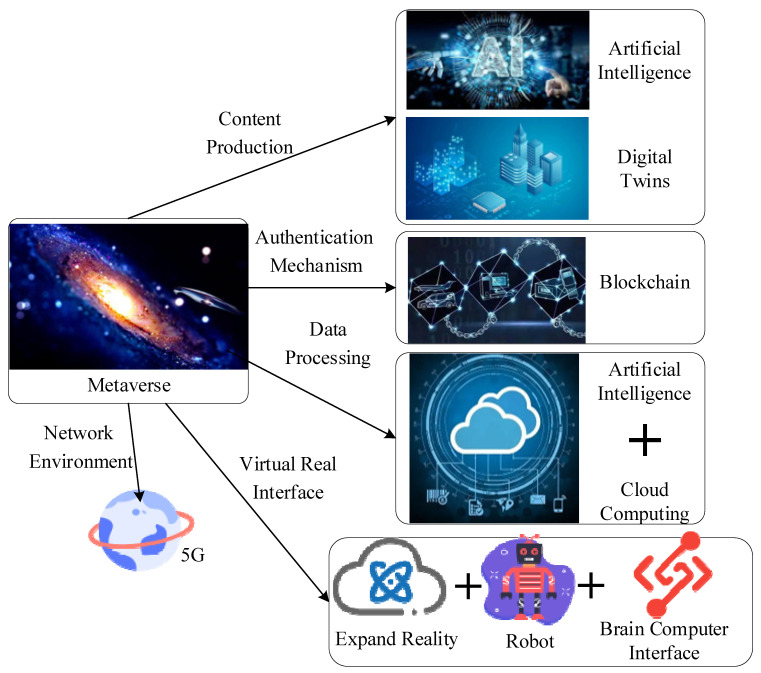
The technical support of the Metaverse.

**Figure 4 sensors-22-09294-f004:**
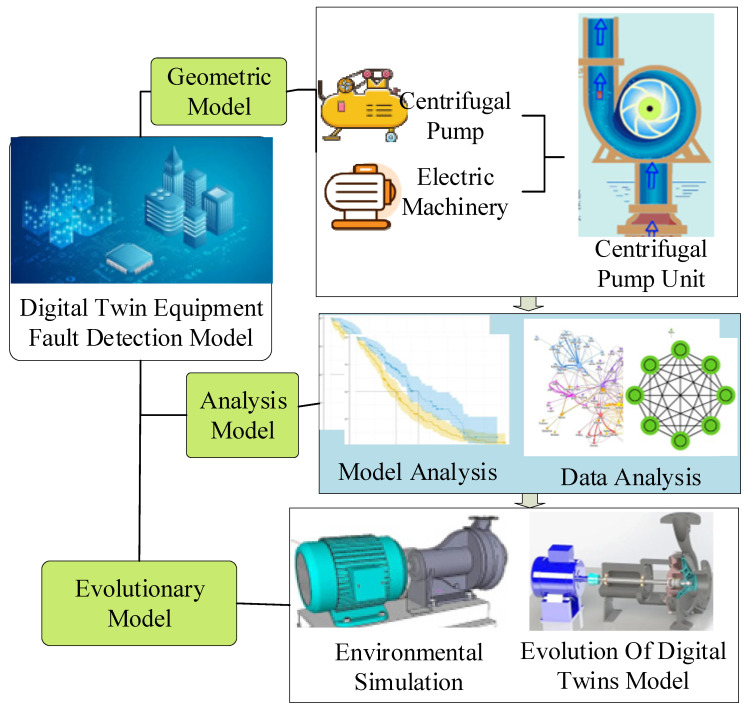
DTT-based applications in centrifugal-pump-oriented fault diagnosis.

**Figure 5 sensors-22-09294-f005:**
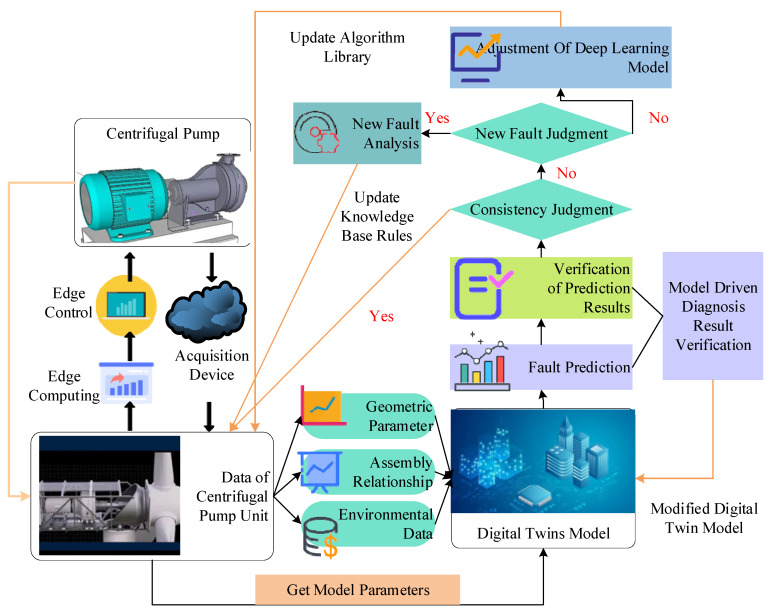
Example of DTT-based IFD’s technical scheme.

**Table 1 sensors-22-09294-t001:** Summary of the main research achievements in digital twins in fluid machinery.

Literature Source	Research Method	Research Contribution	Challenges and Analysis
[33]	DTT + anomaly detection + Bayesian	Continuous abnormal detection of the pump	DTT ensured that assets retained their original intended functionality throughout their life cycle.
[34]	DTT framework	Generalized learning system model	The existing chiller model could not be updated in real time, which was unsuitable for real-time interaction between digital twins and the real physical system.
[39]	DTT + Monte Carlo method	Excellent diagnostic performance of the pump	Through digital twin simulation training, the fault detection, isolation, and quantification of pump-type hydraulic units could be completed.
[40]	DTT + deep transfer learning	IFD framework of mechanical system based on DTT and deep transfer learning	The digital twins of the machine could realize IFD under a limited amount of measured data.
[42]	Expert rules + DTT + machine learning	Remote diagnosis algorithm	Oil wells equipped with submersible electric pumps and sucker rod pumps were brought into stable production to reduce the number of pump stops and failures.
[52]	CFD + automotive electronic pump	An intelligent optimization model	The development of CFD technology provided an opportunity to achieve optimal fluid machinery design in a limited time.
[53]	Removable design + AR	Two virtual buttons added to AR	This helps the operator find the best disassembly sequence for the hydraulic pump in terms of time and cost.
[58]	Noise reduction of centrifugal pump + bionic transformation of fans	Analysis of the influence of two different bionic structures on the structure of a centrifugal pump	The noise reduction capability of the sawtooth structure was not suitable for the high-frequency band.
[59]	Booster pump system + sensor	Turbine flow sensor	A low-flow pump saved more power.
[78]	Fuzzy rules + DTT	Good robustness of the digital twin model	The digital twin model of the whole plant enabled experts to safely test the impact of parameter changes on the process.
[86]	DTT + fan blade repair	Digital twins developed for the grinding process	The computer vision system could track the state of the fan blades during the repair process.
[89]	DTT	Proposed solutions to existing challenging problems	The DTT framework enabled real-time monitoring, fault diagnosis, and operation optimization of the OWT support structure.
[95]	Vortex distribution method + low-pressure axial-flow fan	CFD simulation strategy	Optimizing the aerodynamic performance of low-pressure axial-flow fans with a small tip diameter ratio was important.
[97]	Large eddy current simulation model + axial-flow fan aerodynamics	Proposed new blades	A serrated trailing-edge blade could reduce the aerodynamic noise of an axial-flow fan.
[98]	Aerodynamics + cross-flow fan loss model	Modeling and experiments helping infer the performance of cross-flow fans in turbomachinery	The high efficiency and performance of cross-flow fans depended on the design parameters to a large extent.
[100]	Diagonal-flow fan + Pearson correlation analysis + CFD	Effective optimization of the robustness of corner-flow fans	The combination of Pearson correlation analysis, the CFD calculation method, and the Kriging agent model could be applied to the performance optimization of different types of fans.

**Table 2 sensors-22-09294-t002:** Comparison of main research results between this study and other similar published literature.

Reference Number	Research Type	Research Contribution	Research Conclusions
[105]	Review	The authors investigate the key technologies of a blockchain in the Metaverse and demonstrate the role of a blockchain in Metaverse applications and services.	The Metaverse has yet to prove it can secure its users’ digital content and data.
[106]	Review/methodology	The authors provide a three-tier architecture that links the Metaverse and the physical world.	Using DTT in the Metaverse has security and privacy challenges.
[107]	Methodology	The authors introduce a digital twin framework for evaluating the health of mechanical systems.	It is proved that DTT can reliably evaluate mechanical system health.
[108]	Review	The author reviews the technologies and tools that support digital twins.	DTT is far from reaching its potential, and it is a complex system and a long process.
[109]	Review	The authors conduct the most advanced investigation of DTT.	The modular digital twins, the consistency and accuracy of modeling, and the incorporation of big data analysis into the twin model are verified.
This study	Review	The author investigates the application of DTT and the Metaverse in fluid machinery.	The application of DTT in fan and pump fluid machinery is mature, while Metaverse cases are few.

## Data Availability

Not applicable.

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
