# Peer review of "Application of Digital Twins and Metaverse in the Field of Fluid Machinery Pumps and Fans: A Review"

_sensors, 2022, doi:10.3390/s22239294_

Round 1

Reviewer 1 Report

This paper proposed a review paper on the application of digital twins and metaverse in the field of fluid machinery pumps and fans. In general, this review paper has certain effect in this area, and some comments are listed as follows:

1. The layout of figures is suggested to be adjusted. For example, the size of picture 2 is inconsistent, and the lines of picture 3 are cumbersome.

2.  The research background of digital twin in Part 2.2.1 is suggested to be replenished to the introduction.

3. The serial number of the tertiary heading in line 154 of this paper is incorrect and the author should change.

4. Part 2.3 studying mechaninery pumps from the perspective of digital twin and  metaverse does not mention the analytical relationship.

5. In figure 5, the description of data-driven fault diagnosis method for real-time prediction of vibration signal of centrifugal pump unit is too confusing and not accurately expressed.

6. The content of 2.3.2 includes both methods and results, whereas the title of the tertiary heading deals only with results.

7. The summary and analysis section of Table 1 and 2 suggests a concise overview and an overly wordy presentation.

Author Response

This paper proposed a review paper on the application of digital twins and metaverse in the field of fluid machinery pumps and fans. In general, this review paper has certain effect in this area, and some comments are listed as follows:
1. The layout of figures is suggested to be adjusted. For example, the size of picture 2 is inconsistent, and the lines of picture 3 are cumbersome.

Reply: Thank you for your comments after careful reading. Figure 2 and Figure 3 have been adjusted. The size of Figure 2 has been adjusted to be consistent, and the lines in Figure 3 have been simplified.

2.  The research background of digital twin in Part 2.2.1 is suggested to be replenished to the introduction.

Reply: Thank you for your comments after careful reading. The introduction has been revised. The contents related to the digital twins background in Section 2.2.1 are added to the introduction section, which makes the introduction section more complete.

3. The serial number of the tertiary heading in line 154 of this paper is incorrect and the author should change.

Reply: Thank you for your comments after careful reading. The title number of 154 lines has been modified, and the number has been changed to 2.2.2.

4. Part 2.3 studying mechaninery pumps from the perspective of digital twin and  metaverse does not mention the analytical relationship.

Reply: Thank you for your careful reading and suggestions. Section 2.3 has been revised. At the beginning of Section 2.3, the application description of digital twins and metaverse in mechanical pumps has been added, emphasizing the relationship between digital twins and metaverse and mechanical pumps.

5. In figure 5, the description of data-driven fault diagnosis method for real-time prediction of vibration signal of centrifugal pump unit is too confusing and not accurately expressed.

Reply: Thank you for your careful reading, affirmation and suggestions. Figure 5 has been optimized, the overall structure of Figure 5 has been adjusted, the links between various elements have been added, unnecessary content has been simplified, and the process of fault diagnosis has been highlighted.

6. The content of 2.3.2 includes both methods and results, whereas the title of the tertiary heading deals only with results.

Reply: Thank you for your careful reading and suggestions. Therefore, the title of Section 2.3.2 has been modified. The original: "Research Results of Fluid Mechanical Pump from the Perspective of Metaverse" has been revised to "Application and Research Results of the Metaverse in Fluid Machinery".

7. The summary and analysis section of Table 1 and 2 suggests a concise overview and an overly wordy presentation.

Reply: Thank you for your careful reading and suggestions. Therefore, Table 1 and Table 2 have been modified. The original contents of Table 1 and Table 2 have been simplified and merged into Table 1 to summarize the methods, contributions, problems, and challenges proposed in the research literature in Sections 2.3 and 2.4 above.

Reviewer 2 Report

- There are some minor grammatical errors and tyops. For example "The axial flow Fan can significantly 24 improve aerodynamic performance..." Should have been "The axial flow fan....." Proofread the article carefully.

- The current review has to be compared with recent reviews on digital twins and metaverse such as Gadekallu, T. R., Huynh-The, T., Wang, W., Yenduri, G., Ranaweera, P., Pham, Q. V., ... & Liyanage, M. (2022). Blockchain for the Metaverse: A Review. arXiv preprint arXiv:2203.09738.

- WHat are the key contributions of this study to the literature?

- Section 2.3 and 2.4 have to be summarized in a table with the attributes such as contribution, challenges for each of the related work.

- Even though DTT and metaverse are an exciting and interesting solutions, there are at their infancy. There are several challenges that need to be resolved to realize their full potential. A section challenges, potential solutions and future directions has to be added.

- WHat are the limitations of this study?

Author Response

There are some minor grammatical errors and tyops. For example "The axial flow Fan can significantly 24 improve aerodynamic performance..." Should have been "The axial flow fan....." Proofread the article carefully.

Reply: Thank you for your careful reading and suggestions. For the grammar and errors in the translation, professional proofreaders who are native English speakers were invited to proofread the whole manuscript.

- The current review has to be compared with recent reviews on digital twins and metaverse such as Gadekallu, T. R., Huynh-The, T., Wang, W., Yenduri, G., Ranaweera, P., Pham, Q. V., ... & Liyanage, M. (2022). Blockchain for the Metaverse: A Review. arXiv preprint arXiv:2203.09738.
Reply: Thank you for your careful reading and suggestions. For the comparison part you mentioned, this revision is compared with these reviews on digital twins and the metaverse in the discussion part of Section 4. The newly added document number is [101] and [102]. Through comparison, we find that the research in this paper has similar conclusions with the similar research, indicating that the comments in this paper are of practical value and theoretical reference significance.

- WHat are the key contributions of this study to the literature?
Reply: Thank you for your question. This research starts with the development of digital twins and the metaverse, mainly studies the current situation of their application in fluid mechanical pumps and fans, and makes a summary, analysis, and discussion. Then, it has important theoretical value and reference significance for the current similar research literature on the limitations and future development direction of digital twins and the metaverse.

- Section 2.3 and 2.4 have to be summarized in a table with the attributes such as contribution, challenges for each of the related work.
Reply: Thank you for your careful reading and suggestions. The results of Section 2.3 and Section 2.4 are described in Table 1 and Table 2, respectively. The original contents in Table 1 and Table 2 have been simplified and consolidated into Table 1, which is used to summarize the methods, contributions, problems, and challenges proposed by the research literature above Sections 2.3 and 2.4.

- Even though DTT and metaverse are an exciting and interesting solutions, there are at their infancy. There are several challenges that need to be resolved to realize their full potential. A section challenges, potential solutions and future directions has to be added.

Reply: Thank you for your suggestion. Section 5 has been added to the text to describe the challenges, potential solutions, and future development direction of Digital Twins and Metaverse in the industrial field, especially in the direction of fluid machinery.

- WHat are the limitations of this study?

Reply: Thank you for your question. A description of the research limitations has been added in the conclusion section. The limitation of this study is that the number of literatures reviewed is limited. The current situation of pumps and fans in fluid machinery has only been investigated, but not in the whole fluid machinery field. It is necessary to further screen the newly published digital twins and metaverse related literatures in future work and make a further analysis.

Reviewer 3 Report

  This is a very well written article and I enjoyed it. However, I am not sure if this paper is classified as a "review". Because the structure of the essay is standard for a research paper, I think it might be considered a research paper using literature review as the method.

  The content of the article is impeccable, I think the first two sections are well written, but the last three sections are a little too brief than the first two.

  My suggestion is therefore to expand the content of the last three sections, especially the discussion section, where I would like to see more relevant discussions on the integration of the digital twin with the industrial sector.

Author Response

Reviewer 3

This is a very well written article and I enjoyed it. However, I am not sure if this paper is classified as a "review". Because the structure of the essay is standard for a research paper, I think it might be considered a research paper using literature review as the method.
Reply: Thank you for your careful reading and affirmation. The structure and organization of the paper does contain some technical content. Therefore, this revision will be based on your opinion, and will be revised to a research paper that is more inclined to take the literature review as the method. The introduction section specifically adds the background description of digital twins, and further improves the overall image framework and quality, and then adds the close connection between digital twins and industry for the following discussion.

  The content of the article is impeccable, I think the first two sections are well written, but the last three sections are a little too brief than the first two.
Reply: Thank you for your careful reading, affirmation, and suggestions. The structure and organization of the paper really need to be adjusted. Therefore, according to your opinions, this revision added the close connection between digital twins and industry to the following three parts, especially the discussion part, and expand the results part, combine the contents of Table 1 and Table 2, and add the description and analysis of the summary part to make the second half fuller.

  My suggestion is therefore to expand the content of the last three sections, especially the discussion section, where I would like to see more relevant discussions on the integration of the digital twin with the industrial sector.

Reply: Thank you for your comments after careful reading. This revision expands the contents of the three parts from Section 3 to Section 6. It adds a description of the application of digital twins and the metaverse in mechanical pumps and their relationship. It also discusses the integration of industrial fields and different parts. The industrial digital twins platform is a platform that can realize elastic expansion, equipment access, equipment operation, and maintenance. The integrated platform of digital factory is of great significance to the realization of digital, visual, and intelligent management and operation of industry.

Reviewer 4 Report

The paper describes a review of the application of digital twins and metaverse in the field of pumps and fans for fluid machinery. The paper describes a review of the application of digital twins and metaverses in the field of pumps and fans for fluid machinery. The abstract does not clearly indicate the authors' contributions to the study problem addressed. Some of these details appear throughout the document. In this way, the advances, results, conclusions, and importance of the research carried out are not well appreciated. It is advisable to rewrite the summary indicating the main contributions, as well as the conclusions obtained.

The document proposes a revision. It is advisable to show the main research studies related to this topic, as well as other problems associated with the problem addressed. It may be interesting to make comparisons with other research works already published, showing the techniques used by other authors. In this case, the main contributions of the authors to the problem analyzed have not been very clear, nor have the advantages and disadvantages compared to other research carried out. As a suggestion, perhaps the elaboration of some additional comparative table that compiles the main contributions to the study carried out, incorporating the information from the bibliographical references, can give more clarity to the state of the art. The number of bibliographical references cited in the document is adequate for the study carried out.

On the other hand, the document is technically solid. The concepts have been presented comprehensively. The different figures, tables, diagrams, and schemes attached facilitate the understanding of the contents described by the authors in this document. The only drawback is that some figures are so small that it is impossible to see all the details inside. However, the information provided supports the comments and ideas made by the authors. All this makes it easier for the reader to follow the document.

Most of the text incorporated in the conclusions section is more geared towards a discussion section than the conclusions themselves. It would be advisable to rewrite this section and include, if the authors consider it necessary, a section for the discussion of results. The conclusions section should include the main ideas, comments, contributions, and results obtained by the authors during the study.

Author Response

The paper describes a review of the application of digital twins and metaverse in the field of pumps and fans for fluid machinery. The paper describes a review of the application of digital twins and metaverses in the field of pumps and fans for fluid machinery. The abstract does not clearly indicate the authors' contributions to the study problem addressed. Some of these details appear throughout the document. In this way, the advances, results, conclusions, and importance of the research carried out are not well appreciated. It is advisable to rewrite the summary indicating the main contributions, as well as the conclusions obtained.
Reply: Thank you for your comments. After careful reading, we have revised the abstract, reorganized the results and conclusions in the abstract, and explained the research progress, results, and conclusions of digital twins and metaverse in fluid machinery, as well as the importance of this research, making the description of the abstract clearer.

The document proposes a revision. It is advisable to show the main research studies related to this topic, as well as other problems associated with the problem addressed. It may be interesting to make comparisons with other research works already published, showing the techniques used by other authors. In this case, the main contributions of the authors to the problem analyzed have not been very clear, nor have the advantages and disadvantages compared to other research carried out. As a suggestion, perhaps the elaboration of some additional comparative table that compiles the main contributions to the study carried out, incorporating the information from the bibliographical references, can give more clarity to the state of the art. The number of bibliographical references cited in the document is adequate for the study carried out.
Reply: This is a good suggestion. We have developed an additional comparison table (Table 2) in this revision. In this table, we listed and compared the installation literature of the same kind of research. In addition to the original literature, we also added four references 107-109 for content comparison, including the methods, conclusions, limitations, etc. used in this study and other studies. In this way, we can clearly see the practical value of this study and the disadvantages that need to be improved, and provide direction for future research.

On the other hand, the document is technically solid. The concepts have been presented comprehensively. The different figures, tables, diagrams, and schemes attached facilitate the understanding of the contents described by the authors in this document. The only drawback is that some figures are so small that it is impossible to see all the details inside. However, the information provided supports the comments and ideas made by the authors. All this makes it easier for the reader to follow the document.
Reply: Thank you for your affirmation and suggestions after careful reading. It is true that figures and tables can help readers understand the content of the text more clearly. Therefore, this revision has enlarged and adjusted the relevant elements in all images to make the pictures clearer. Table 2 has also been added to compare similar studies to highlight the advantages and disadvantages of this study and help readers better understand the article.

Most of the text incorporated in the conclusions section is more geared towards a discussion section than the conclusions themselves. It would be advisable to rewrite this section and include, if the authors consider it necessary, a section for the discussion of results. The conclusions section should include the main ideas, comments, contributions, and results obtained by the authors during the study.

Reply: Thank you for your affirmation and suggestions after careful reading. The conclusion has been modified as required. We moved some elements of the previous version of the findings to Section 4 and reorganized the conclusions according to the main ideas, observations, contributions, and results of the research process.

Round 2

Reviewer 1 Report

The authors have addressed all my comments.

Reviewer 2 Report

The authors have addressed all the comments. I have no further comments / queries